# Causal Reasoning with Large Foundation Models to Guide Dynamic System Forecasting

## Abstract

Prevailing data-driven models for spatio-temporal forecasting excel at interpolating within known patterns but often falter in critical real-world scenarios. This failure stems from a fundamental flaw: they learn **spurious correlations** from raw data, bypassing the underlying semantic and physical principles that form the true causal pathways. To address this, we introduce `PRISM` (Physics-informed Reasoning and Interpretation for Spatio-temporal Modeling), a framework that performs a *principled causal intervention*. `PRISM` employs a Vision-Language Model (VLM) to interpret spatio-temporal snapshots into semantic narratives, and a Large Language Model (LLM) to reason with these narratives and explicit physical laws, generating a causally-informed textual guidance. This guidance is then encoded to steer a downstream numerical predictor. Extensive experiments across fluid dynamics, weather forecasting, and urban traffic demonstrate that this intervention significantly enhances model capabilities. By repairing the causal chain, boosts *out-of-distribution (OOD) generalization*, improves *prediction under data sparsity*, and sharpens *extreme event prediction*. Our work propose a new paradigm that unifies the pattern recognition of traditional models with the causal reasoning of large foundation models, paving the way for more reliable AI in science. Codes are available at https://anonymous.4open.science/status/PRISM-8BF5.

## 1 Intruction

The modeling and prediction of dynamical systems serve as a cornerstone of modern science and engineering (Li et al., 2022; 2021; Kochkov et al., 2021), exerting a profound influence across diverse domains ranging from weather forecasting (Bi et al., 2023; Wu et al., 2025; Gao et al., 2025; Lam et al., 2023) and climate science (Gentine et al., 2021; Bordoni et al., 2025) to urban traffic (Wang et al., 2020; Wu et al., 2024a) management and fluid dynamics analysis (Wu et al., 2024b;c). The ability to accurately foresee the future states of these complex systems is crucial to advance scientific discovery, mitigate disasters, optimize resources, and enable effective decision making. In recent years, the machine learning community has witnessed a burgeoning development in leveraging data-driven approaches, particularly deep learning, for spatio-temporal prediction (Wu et al., 2023; Shi et al., 2015). These methods demonstrate immense potential for capturing intricate spatial correlations and temporal dependencies directly from observational data.

However, despite these significant advances, applying these data-driven models to real-world scenarios often exposes their fundamental limitations: an "***Achilles heel***" that hinders their practical reliability. ❶ A primary challenge is their fragile out-of-distribution (OOD) generalization (Yang et al., 2024; Ma et al., 2023; Volpi et al., 2018; Kirchmeyer et al., 2022). Trained predominantly on historical data, these models excel at interpolating within seen patterns but tend to suffer sharp performance degradation when confronted with novel scenarios, such as unprecedented atmospheric blocking events (Wang et al.) or anomalous traffic congestion triggered by unforeseen incidents (Wu et al., 2024a). ❷ Furthermore, real-world observational data are frequently sparse, incomplete, or noisy due to sensor limitations, network failures, or occlusions (Luo et al., 2024; Lienen & Günnemann, 2022;?). Standard models often struggle to provide reliable predictions under such data scarcity, as they lack mechanisms for robust contextual inference or for leveraging physical constraints to intelligently fill information gaps. ❸ Compounding the problem is the inherent difficulty in predicting extreme events (Shu et al., 2025b; Zhang et al., 2023; Shu et al., 2025a; Wu et al., 2025). These low-probability, high-impact events are sparsely represented in historical datasets, making it challenging

for purely data-driven models to learn their triggering mechanisms or accurately identify subtle precursor signals that domain experts might readily recognize.

These persistent challenges collectively point to a core deficiency in the current paradigm: the learning process is largely confined to statistical correlations on the data's surface, failing to grasp the deeper principles governing the system's evolution. Specifically, these models face *two fundamental gaps*: a **Semantic Gap**, where they struggle to extract high-level concepts from raw data, and a **Physics Gap**, where they lack an explicit understanding of intrinsic physical laws.

This neglect of semantics and physics can be understood more profoundly from a causal perspective. As illustrated in **Figure 1(a)**, traditional models attempt to learn a direct mapping from historical observations ($X_{\text{hist}}$) to future predictions ($Y_{\text{pred}}$). However, this path bypasses the true drivers of the system's evolution the unobserved latent semantic states ($S$) and the invariant physical laws ($P$). Consequently, the model learns a fragile, superficial **spurious correlation**. *This is the fundamental reason for their frequent failures in OOD scenarios, as this statistical shortcut breaks down when the data distribution shifts.*

To address this core causal flaw, we introduce `PRISM` (Physics-informed Reasoning and Interpretation for Spatiotemporal Modeling), a novel framework designed to perform a **causal intervention**, as depicted in **Figure 1(b)**. Instead of learning the brittle shortcut, `PRISM` actively estimates the latent semantic state $\hat{S}$ via a Vision-Language Model (VLM) (Zhang et al., 2024; Zhou et al., 2022; Guo et al., 2024) and performs causal reasoning by synthesizing it with explicit, textualized physical principles

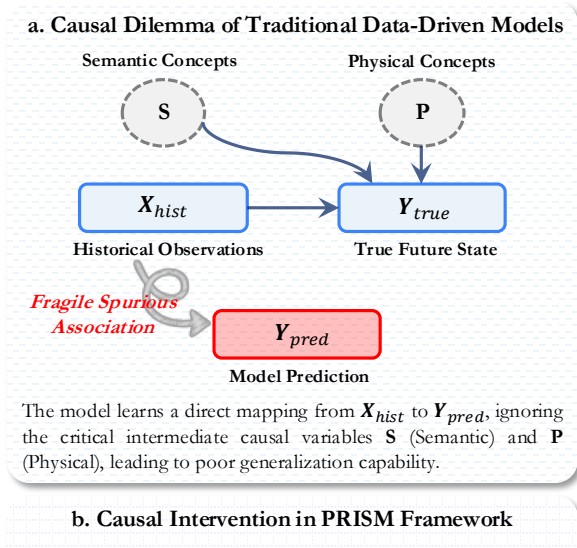

**a. Causal Dilemma of Traditional Data-Driven Models**

The model learns a direct mapping from $X_{hist}$ to $Y_{pred}$, ignoring the critical intermediate causal variables **S** (Semantic) and **P** (Physical), leading to poor generalization capability.

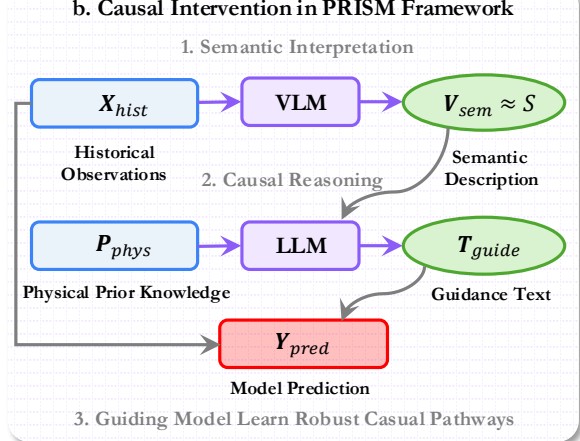

**b. Causal Intervention in PRISM Framework**

Figure 1: **Causal diagrams illustrating the core idea of PRISM. (a)** Traditional models learn a *spurious correlation* (red dashed arrow) by ignoring latent semantic ($S$) and physical ($P$) factors. **(b)** `PRISM` performs a *causal intervention* by using a VLM and LLM to generate a guidance text ($T_{\text{guide}}$) that encapsulates these factors, steering the prediction towards a robust causal path (green arrow).

via a Large Language Model (LLM) (Naveed et al., 2025; Zhao et al., 2023). This process generates a causally-informed guidance text ($T_{\text{guide}}$) containing state analyses and evolutionary trends. Finally, in the **Multi-Modal Fusion** stage, this guidance is encoded and fused with the original data. This augmented input steers any downstream predictor (e.g., ConvLSTM (Shi et al., 2015), Transformer (Vaswani et al., 2017; Dosovitskiy et al., 2021), FNO (Li et al., 2020)) to learn a more robust causal pathway, fundamentally enhancing its generalization and reliability.

The contributions of this paper are summarized as follows: ***Novel Methodology***: We are the first to systematically design a workflow that leverages a VLM and an LLM to perform a causal intervention in spatio-temporal forecasting, bridging the semantic and physical gaps. ***Superior Performance***: Extensive experiments demonstrate that `PRISM`, as a plug-and-play module, significantly boosts model performance, especially in tackling the core challenges of *OOD generalization*, *prediction under data sparsity*, and *extreme event forecasting*. ***A Pioneering Paradigm***: Our work pioneers a

new paradigm for developing more generalizable and reliable AI for science by unifying the pattern recognition of traditional models with the causal reasoning capabilities of large foundation models.

## 2 RELATED WORK

**Data-Driven Dynamical System Modeling.** Modeling complex dynamical systems has been a focal point of deep learning research. A diverse array of architectures, including Convolutional Neural Networks (CNNs) (Gao et al., 2022a), Recurrent Neural Networks (RNNs) (Wang et al., 2022), and Transformers (Gao et al., 2022b), have been developed to capture intricate spatio-temporal dependencies. More recently, Neural Operators (Li et al., 2021; Hao et al., 2023) have emerged as a powerful paradigm for learning mappings between infinite-dimensional function spaces, showing promise in scientific simulations. In a parallel effort, Physics-Informed Neural Networks (PINNs) (Cai et al., 2021) attempt to embed physical knowledge by incorporating governing equations as soft constraints during training. However, while these methods excel at pattern recognition and interpolation, they often function as "black boxes," struggling to reason about unseen scenarios or explicitly leverage high-level conceptual knowledge, a gap our work aims to fill.

**Out-of-Distribution (OOD) Generalization.** The ability to generalize out-of-distribution (OOD) (Ye et al., 2021; Yang et al., 2024) is a critical frontier for machine learning, ensuring model reliability under real-world distribution shifts. The field has seen rapid progress, with prominent techniques including invariant causal inference (Wu et al., 2024e;c), which seeks robust predictors across environments, and various forms of data augmentation (Bai et al., 2021) and domain adaptation (Ding et al., 2022) to expose models to wider data variations. While effective, many of these approaches operate at the data or feature level. They often lack a mechanism to incorporate external, symbolic knowledge, such as physical laws, which provides a universal basis for generalization. Our framework addresses this by using language to inject such principled knowledge.

**Large Language and Vision-Language Models.** The recent ascendancy of Large Language Models (LLMs) (Chang et al., 2024; Guo et al., 2025) and Multimodal LLMs (MLLMs) (Lin et al., 2023; Bai et al., 2025) has revolutionized AI, endowing models with unprecedented capabilities in reasoning, understanding, and generation across modalities. In the vision-language domain, MLLMs have demonstrated a remarkable ability to interpret and reason about dynamic visual content, moving beyond simple perception tasks (Cheng et al., 2024). To date, the power of these foundation models has been predominantly applied to tasks centered on human language and common-sense reasoning. Their potential to serve as symbolic reasoning engines for complex scientific domains specifically, to interpret physical phenomena and guide numerical forecasting models remains largely untapped. Our work pioneers this novel application, positioning LLMs not as end-point predictors, but as co-reasoning partners in the scientific modeling loop.

## 3 METHODOLOGY

Our work aims to address the causal deficiencies inherent in traditional data-driven models for dynamical system forecasting. To this end, we propose `PRISM` (Physics-informed Reasoning and Interpretation for Spatio-temporal Modeling). This section first formalizes the problem domain and then elaborates on the three core stages of the `PRISM` framework: (1) semantic perception via Vision-Language Models, (2) causal reasoning with physical priors, and (3) guided multi-modal fusion for prediction.

### 3.1 PROBLEM FORMULATION AND `PRISM` FRAMEWORK OVERVIEW

***Dynamical Systems and Observation Space.*** We consider a discrete-time dynamical system defined in a state space $\mathcal{S} \subseteq \mathbb{R}^{D_s}$. The evolution of the system is governed by a latent, potentially stochastic Markovian transition kernel $\mathcal{P}$, where a new state $\mathbf{s}_{t+1} \sim \mathcal{P}(\cdot|\mathbf{s}_{\leq t}, \Theta)$. Here, $\mathbf{s}_{\leq t} = (\mathbf{s}_0, \ldots, \mathbf{s}_t)$ denotes the historical state trajectory, and $\Theta$ represents the intrinsic parameters of the system. We do not directly access the true state $\mathbf{s}_t$; instead, we obtain projections through an observation operator $\mathcal{O} : \mathcal{S} \to \mathcal{X}$, which yield observations $\mathbf{x}_t = \mathcal{O}(\mathbf{s}_t)$ in an observation space $\mathcal{X} \subseteq \mathbb{R}^{H \times W \times C}$.

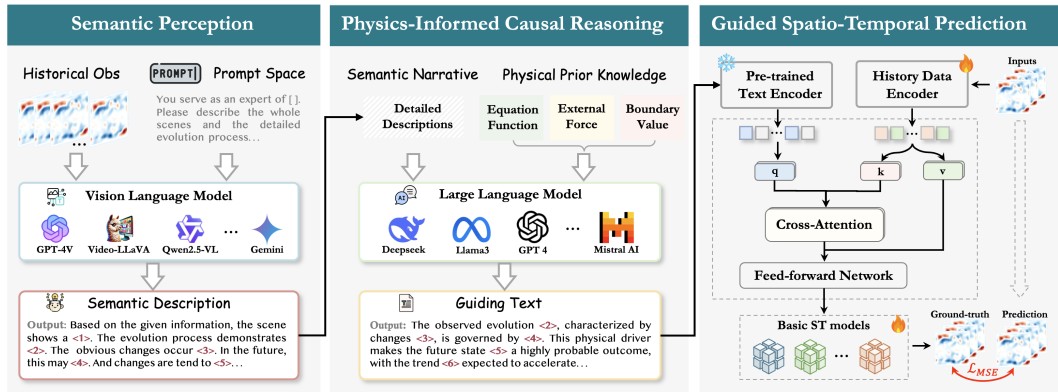

Figure 2: **The Architecture of the PRISM Framework.** The framework consists of three core stages to perform a causal intervention in forecasting. **(1)** *Semantic Perception*: A Vision-Language Model (VLM) interprets raw historical observations and, guided by a prompt, generates a high-level *Semantic Narrative*. **(2)** *Physics-Informed Causal Reasoning*: A Large Language Model (LLM) synthesizes this narrative with explicit *Physical Prior Knowledge* (e.g., governing equations) to produce a predictive and causally-informed *Guiding Text*. **(3)** *Guided Spatio-Temporal Prediction*: The guiding text is encoded into a vector, which is then fused with numerical history data via a cross-attention mechanism. This fused information steers a downstream spatio-temporal (ST) model to generate a more robust and physically consistent forecast.

***The Conventional Forecasting Paradigm and its Causal Limitations.*** Given a history of $P$ observations, $\mathbf{X}_{\text{hist}}^{(P)} = (\mathbf{x}_{t-P+1}, \ldots, \mathbf{x}_t)$, the forecasting task is to estimate the future sequence of $K$ observations, $\mathbf{X}_{\text{future}}^{(K)} = (\mathbf{x}_{t+1}, \ldots, \mathbf{x}_{t+K})$. Conventional data-driven approaches seek to learn a parameterized function $f_\theta : \mathcal{X}^P \to \mathcal{X}^K$ by minimizing the expected loss over a data distribution $\mathcal{D}$:

$$\theta^* = \arg \min_\theta \mathbb{E}_{(\mathbf{X}_{\text{hist}}^{(P)}, \mathbf{X}_{\text{future}}^{(K)}) \sim \mathcal{D}} \left[ \mathcal{L}(f_\theta(\mathbf{X}_{\text{hist}}^{(P)}), \mathbf{X}_{\text{future}}^{(K)}) \right] \tag{1}$$

As illustrated by the causal diagram in Figure 1(a), this paradigm learns a direct associative path from $\mathbf{X}_{\text{hist}}$ to $\mathbf{X}_{\text{future}}$, bypassing the true causal chain mediated by latent **semantic structures** $S_t$ and invariant **physical principles** $\mathcal{P}_{\text{phys}}$.

***PRISM's Guided Forecasting Paradigm.*** PRISM performs a causal intervention by introducing an externally generated **guidance variable** $\mathbf{g}$. We aim to learn an enhanced prediction model $f_\phi^* : \mathcal{X}^P \times \mathbb{R}^{d_g} \to \mathcal{X}^K$, which conditions on both the historical observations and this guidance variable. The guidance variable $\mathbf{g}$ is the output of a sophisticated generation process $\mathcal{G}$, designed to capture the omitted causal information:

$$\mathbf{g} = \mathcal{G}(\mathbf{X}_{\text{hist}}^{(P)}, \mathcal{P}_{\text{phys}}; \mathcal{M}_{\text{VLM}}, \mathcal{M}_{\text{LLM}}, \mathcal{E}_{\text{text}}) \tag{2}$$

where $\mathcal{P}_{\text{phys}}$ is a repository of physical knowledge. The enhanced prediction problem is thus reformulated as optimizing the parameters $\phi$ of the model $f_\phi^*$:

$$\phi^* = \arg \min_\phi \mathbb{E}_{(\mathbf{X}_{\text{hist}}^{(P)}, \mathbf{X}_{\text{future}}^{(K)}) \sim \mathcal{D}} \left[ \mathcal{L}(f_\phi^*(\mathbf{X}_{\text{hist}}^{(P)}, \mathbf{g}), \mathbf{X}_{\text{future}}^{(K)}) \right] \tag{3}$$

### 3.2 STAGE 1: FROM OBSERVATIONS TO SEMANTIC NARRATIVES

This initial stage aims to decode a high-quality estimate of the latent semantic state $S_t$ from the high-dimensional observations $\mathbf{X}_{\text{hist}}^{(P)}$ using a large Vision-Language Model (VLM), $\mathcal{M}_{\text{VLM}}$.

We define the entire VLM operation as a **semantic perception operator**, $\Psi_{\text{sem}}$. First, a visualization map $\mathcal{V} : \mathcal{X}^P \to \mathcal{I}^P$ renders the numerical grids into a sequence of images, $\mathbf{I}_{\text{hist}} = \mathcal{V}(\mathbf{X}_{\text{hist}}^{(P)})$, where $\mathcal{I}$ is the image space. Subsequently, a carefully designed **prompting function** $\Pi_{\text{sem}} : \mathcal{I}^P \to \mathcal{T}$ wraps

this image sequence into a structured query, where $\mathcal{T}$ is the text space. The VLM then generates the semantic narrative $T_{\text{sem}}$ based on this prompt:

$$T_{\text{sem}} = \Psi_{\text{sem}}(\mathbf{X}_{\text{hist}}^{(P)}) \triangleq \mathcal{M}_{\text{VLM}}(\Pi_{\text{sem}}(\mathcal{V}(\mathbf{X}_{\text{hist}}^{(P)}))) \tag{4}$$

Here, $T_{\text{sem}}$ is a natural language text that serves as a concrete instantiation and rich description of the abstract semantic variable $S_t$.

### 3.3 STAGE 2: CAUSAL REASONING UNDER PHYSICAL CONSTRAINTS

This stage features a **causal reasoning operator**, $\Psi_{\text{reason}}$, implemented by a Large Language Model (LLM), $\mathcal{M}_{\text{LLM}}$. It is designed to synthesize the observational evidence ($T_{\text{sem}}$) with universal physical principles ($\mathcal{P}_{\text{phys}}$) to infer future outcomes.

Let $\mathcal{P}_{\text{phys}} = \{p_i\}_{i=1}^N$ be a set of $N$ textualized physical laws. We model the LLM's reasoning process as generating a probabilistic description of future states, conditioned on the given evidence and axioms. The input to the LLM is constructed by a **reasoning prompt function** $\Pi_{\text{reason}} : \mathcal{T} \times \mathcal{P}(\mathcal{T}) \to \mathcal{T}$, where $\mathcal{P}(\mathcal{T})$ denotes the power set of text sets. The final guidance text, $T_{\text{guide}}$, is then generated as follows:

$$T_{\text{guide}} = \Psi_{\text{reason}}(T_{\text{sem}}, \mathcal{P}_{\text{phys}}) \triangleq \mathcal{M}_{\text{LLM}}(\Pi_{\text{reason}}(T_{\text{sem}}, \mathcal{P}_{\text{phys}})) \tag{5}$$

Crucially, $T_{\text{guide}}$ transcends simple description, providing instead causal explanations and forward-looking judgments that offer insights beyond pure statistical patterns.

### 3.4 STAGE 3: GUIDED MULTI-MODAL FUSION AND PREDICTION

The final stage is responsible for effectively fusing the symbolic guidance information ($T_{\text{guide}}$) with the sub-symbolic numerical data ($\mathbf{X}_{\text{hist}}^{(P)}$). First, a pre-trained text encoder $\mathcal{E}_{\text{text}} : \mathcal{T} \to \mathbb{R}^{d_g}$ maps the guidance text into a dense guidance vector $\mathbf{g} \in \mathbb{R}^{d_g}$. Concurrently, a data encoder $\mathcal{E}_{\text{data}} : \mathcal{X}^P \to (\mathbb{R}^{d_h})^P$ processes the historical observations into a sequence of spatio-temporal features $\mathbf{H}_{\text{hist}} = (\mathbf{h}_1, \ldots, \mathbf{h}_P)$.

We then introduce a **Guidance-Attention Module**, $\mathcal{A}_{\text{guide}}$, to perform the fusion. This module employs a cross-attention mechanism where $\mathbf{g}$ serves as the Query, and $\mathbf{H}_{\text{hist}}$ serves as the Keys and Values, to compute a guided context representation $\mathbf{c}_{\text{guided}}$. Let $\mathbf{h}_i \in \mathbb{R}^{d_h}$ be the feature at timestep $i$. The fusion process is detailed as:

$$\alpha_i = \frac{\exp\left(\frac{(\mathbf{g}\mathbf{W}_Q)(\mathbf{h}_i\mathbf{W}_K)^T}{\sqrt{d_k}}\right)}{\sum_{j=1}^P \exp\left(\frac{(\mathbf{g}\mathbf{W}_Q)(\mathbf{h}_j\mathbf{W}_K)^T}{\sqrt{d_k}}\right)}, \quad \mathbf{c}_{\text{guided}} = \sum_{i=1}^P \alpha_i(\mathbf{h}_i\mathbf{W}_V) \tag{6}$$

where $\mathbf{W}_Q \in \mathbb{R}^{d_g \times d_k}$, $\mathbf{W}_K \in \mathbb{R}^{d_h \times d_k}$, and $\mathbf{W}_V \in \mathbb{R}^{d_h \times d_v}$ are learnable projection matrices. The attention weight $\alpha_i$ is interpreted as the importance of the $i$-th historical timestep for future prediction, as determined by the guidance signal $\mathbf{g}$.

Finally, the prediction is generated by the decoder of the downstream model, $\mathcal{D}_\phi$. This decoder, which may consist of components like feed-forward networks and recurrent or attention-based layers, takes both the original features $\mathbf{H}_{\text{hist}}$ and the guided context $\mathbf{c}_{\text{guided}}$ as input:

$$\hat{\mathbf{X}}_{\text{future}}^{(K)} = \mathcal{D}_\phi(\mathbf{H}_{\text{hist}}, \mathbf{c}_{\text{guided}}) \tag{7}$$

During training, we minimize the loss function $\mathcal{L}(\phi)$, while the parameters of all large foundation models ($\mathcal{M}_{\text{VLM}}, \mathcal{M}_{\text{LLM}}, \mathcal{E}_{\text{text}}$) remain **frozen**. This **parameter-efficient transfer learning** paradigm allows `PRISM` to be integrated into existing architectures in a lightweight manner.

## 4 EXPERIMENTS

To comprehensively evaluate the effectiveness and versatility of our proposed `PRISM` framework, we conduct a series of extensive experiments on three benchmark datasets spanning diverse domains: urban traffic, fluid dynamics, and atmospheric science.

***Datasets and Evaluation Metrics*** Our experiments are based on three representative public datasets. TaxiBJ+ (Wu et al., 2024a) is a widely-used benchmark for urban spatio-temporal forecasting, recording taxi traffic flow in Beijing. The data is processed into a spatial grid of $32 \times 32$, and the task is to predict the traffic flow for the next 2 hours based on observations from the past 4 hours. Navier-Stokes (NS) (Li et al., 2021) is a classic fluid dynamics simulation dataset that models the vorticity evolution of a 2D incompressible fluid. It features a spatial resolution of $64 \times 64$, and the task involves forecasting the next 10 timesteps of the flow field given the previous 10 timesteps. SEVIR (Veillette et al., 2020) is a large-scale meteorological radar dataset focusing on convective storm events. We use its Vertically Integrated Liquid (VIL) data at a $128 \times 128$ resolution. The task is to predict storm evolution over the next hour based on observations from the past hour.

To assess performance from multiple perspectives, we employ a variety of evaluation metrics. The Mean Squared Error (MSE) is used across all tasks to measure pixel-level accuracy. For TaxiBJ+, we additionally use the Structural Similarity Index (SSIM) to evaluate the visual fidelity of the predicted traffic patterns. For Navier-Stokes, the Mean Absolute Percentage Error (MAPE) is employed to assess the relative error. For SEVIR, we use the Critical Success Index (CSI) to precisely measure the model's ability to forecast storm regions of varying intensities.

***Baselines and Implementation Details*** To validate the plug-and-play nature of `PRISM`, we select ten powerful models covering four mainstream architectural families as downstream predictors (backbones): **CNN-based** (ResNet (He et al., 2016), U-Net (Ronneberger et al., 2015)), **Transformer-based** (ViT (Dosovitskiy et al., 2021), SWIN Liu et al. (2021)), **recurrent spatio-temporal networks** (SimVP (Gao et al., 2022a), PastNet (Wu et al., 2024d)), and **neural operators** (Fourier Neural Operator - FNO, Convolutional Neural Operator - CNO (Raonic et al., 2023), U-shaped Neural Operator - UNO (Rahman et al., 2022)).

In our implementation, the `PRISM` framework, by default, utilizes `GPT-4V` for semantic perception and `GPT-4 Turbo` for causal reasoning. The resulting guiding text is encoded into a 768-dimensional vector using a pre-trained `BERT-base` encoder and is fused with the numerical features of the downstream model via a standard cross-attention layer. All models are trained using the `AdamW` optimizer with an initial learning rate of $1e{-}3$ and a cosine annealing schedule. A key aspect of our training strategy is that the parameters of the VLM, LLM, and text encoder within `PRISM` **remain frozen**. We only train the fusion module and the downstream predictor, enabling parameter-efficient transfer learning and lightweight integration.

***Experimental Environment*** All experiments are conducted on a server equipped with four NVIDIA A100 (80GB) GPUs. Our implementation is based on the PyTorch framework and leverages the Hugging Face `transformers` library for interacting with large models. The experimental platform is Ubuntu 20.04 with CUDA version 11.7.

### 4.1 Analysis of Main Results

**Consistent and Significant Performance Improvement.** As demonstrated in Table 1, the `PRISM` framework delivers consistent and significant performance improvements across diverse models and domains. As a plug-and-play module, its effectiveness is validated by numerous strong results. For instance, in urban traffic forecasting (**TaxiBJ+**), `PRISM` enables U-Net to achieve a remarkable **13.1%** reduction in MSE (from 0.109 to 0.095), showcasing its ability to compensate for the semantic limitations of traditional CNNs. In the highly challenging meteorological forecasting task (**SEVIR**), its value is even more prominent. When augmenting the powerful SWIN Transformer, `PRISM` boosts the critical storm-prediction metric, CSI, by a significant **6.4%** (from 0.392 to 0.417). This result highlights that the causal guidance from `PRISM` transcends pixel-level optimization, enabling models to more accurately predict the evolution of critical physical processes.

**Universal Applicability and Enhancement of SOTA Models.** A core strength of the `PRISM` framework lies in its **cross-architectural universality**. As evidenced in Table 1, `PRISM` consistently boosts the performance of diverse architectures, including CNNs, Transformers, recurrent networks, and neural operators. This versatility is particularly striking in the physics-driven Navier-Stokes task, where `PRISM` exhibits strong synergy with neural operators. It enables the CNO to achieve a remarkable **19.0%** reduction in MSE (from 0.116 to 0.094) and the FNO a **17.0%** reduction (from

Table 1: **Comprehensive performance evaluation across diverse forecasting tasks.** This table compares various baseline models in their original form (Ori) versus their performance when augmented with our `PRISM` framework. The $\Delta$ column quantifies the relative improvement in the primary metric (MSE for TaxiBJ$^+$ and Navier-Stokes, CSI for SEVIR). Best results are in **bold**, and significant improvements are highlighted in green denotes model failure.

| Model | TaxiBJ$^+$ | | | Navier-Stokes | | | SEVIR | | |
|---|---|---|---|---|---|---|---|---|---|
| | MSE $\downarrow$ / SSIM $\uparrow$ | | $\Delta_{MSE}$ (%) | MSE $\downarrow$ / MAPE $\downarrow$ | | $\Delta_{MSE}$ (%) | MSE $\downarrow$ / CSI $\uparrow$ | | $\Delta_{CSI}$ (%) |
| | Ori | + PRISM | | Ori | + PRISM | | Ori | + PRISM | |
| *CNN-Based Architectures* | | | | | | | | | |
| ResNet | 0.101/0.827 | 0.099/0.842 | -2.3 | 0.712/26.1 | 0.696/25.3 | -2.2 | 5.12/0.318 | 5.04/0.332 | +4.4 |
| U-Net | 0.109/0.821 | 0.095/0.858 | -13.1 | 0.129/13.8 | 0.135/12.6 | +4.6 | 4.23/0.351 | 4.18/0.362 | +3.1 |
| *Transformer-Based Architectures* | | | | | | | | | |
| ViT | 0.074/0.848 | 0.075/0.857 | +0.9 | 0.138/14.5 | 0.132/13.8 | -4.3 | 4.01/0.379 | 3.89/0.392 | +3.4 |
| Swin | 0.075/0.872 | 0.072/0.888 | -4.0 | 0.131/15.9 | 0.130/14.6 | -0.8 | 3.81/0.392 | **3.48**/0.417 | **+6.4** |
| *Recurrent Spatio-Temporal Architectures* | | | | | | | | | |
| SimVP | 0.036/0.919 | 0.034/0.925 | -4.8 | 0.095/20.4 | **0.089**/10.1 | **-6.7** | 3.52/0.447 | 3.49/0.461 | +3.1 |
| PastNet | 0.031/**0.935** | 0.031/**0.938** | -0.7 | 0.115/11.9 | 0.114/10.9 | -1.3 | 3.42/0.463 | 3.41/0.465 | +0.4 |
| *Neural Operator Architectures* | | | | | | | | | |
| FNO | 0.052/0.648 | 0.052/0.648 | 0.0 | 0.147/11.8 | 0.122/10.1 | -17.0 | —/— | —/— | — |
| CNO | 0.089/0.860 | 0.086/0.867 | -2.7 | 0.116/**11.4** | 0.094/10.2 | -19.0 | 4.42/0.331 | 4.41/0.335 | +1.2 |
| UNO | 0.051/0.889 | 0.047/0.908 | -8.8 | 0.120/11.5 | 0.130/10.2 | +8.3 | 3.69/0.395 | 3.62/0.404 | +2.3 |

0.147 to 0.122). This suggests that the explicit physical reasoning from `PRISM` is highly compatible with architectures designed to solve PDEs. More importantly, `PRISM` also **enhances state-of-the-art (SOTA) models**. For instance, it elevates SimVP, an already strong baseline on the Navier-Stokes task, to a new SOTA performance by reducing its MSE by **6.7%** to a chart-topping **0.089**. These findings confirm that `PRISM` provides a performance enhancement pathway orthogonal to architectural innovation, effectively advancing the frontiers of existing SOTA models.

## 4.2 Ablation Studies

To investigate the sensitivity of the `PRISM` framework to the choice of foundation models, we conducted a series of ablation studies, with results detailed in Table 2. The experiments clearly show that the performance of `PRISM` is positively correlated with the capabilities of the selected foundation models. The top-performing combination of `GPT-4V` and `GPT-4 Turbo` achieves the best results across all tasks, notably reducing the MAPE on the Navier-Stokes dataset from 20.42% to a remarkable **10.12%**. This validates that higher-quality semantic perception and causal reasoning lead to more significant performance gains. More encouragingly, `PRISM` still demonstrates strong performance when using entirely open-source model combinations, such as `Qwen2.5-VL-Max` + `Llama3-70B`, which reduces the MAPE on Navier-Stokes to 12.01%. This result highlights the robustness and practical value of the `PRISM` framework, proving it is not merely dependent on the sheer power of specific proprietary models but offers an accessible and effective solution for the broader research community.

Further analysis of `PRISM`'s core components confirms the integrity and necessity of its design. First, by comparing the ablation results of different components, we find that a high-quality semantic perception stage is the cornerstone of the framework's performance; even with the strongest reasoning model (`GPT-4 Turbo`), using a weaker perception model (`LLaVA-NeXT-34B`) leads to a noticeable performance drop (e.g., CSI on SEVIR decreases from 0.417 to 0.403). Second, the framework integrity experiment underscores the indispensable nature of the semantic perception stage. When the VLM module is completely removed, and reasoning relies solely on a simple textual description of the numerical data, the performance improvement becomes marginal (e.g., the MAPE on Navier-Stokes remains as high as 18.52%). This provides strong evidence that leveraging a VLM to transform complex visual patterns into rich semantic narratives is a critical and irreplaceable step in successfully bridging the gap between raw data and high-level physical concepts.

Table 2: **Ablation study on the choice of foundation models in the `PRISM` framework.** The performance variation across different VLMs and LLMs highlights the impact of their respective capabilities on the final prediction accuracy. The baseline models (SWIN for SEVIR, SimVP for Navier-Stokes) are evaluated without any `PRISM` guidance.

| MODEL CONFIGURATION | | SEVIR | | NAVIER-STOKES | |
|---|---|---|---|---|---|
| | | MSE (↓) | CSI (↑) | MSE (↓) | MAPE (%) (↓) |
| *Baseline (w/o PRISM)* | | *3.81* | *0.392* | *0.0953* | *20.42* |
| *Best Performance with Proprietary Models* | | | | | |
| ★ GPT-4V | GPT-4 TURBO | **3.48** | **0.417** | **0.0887** | **10.12** |
| ★ GEMINI 1.5 PRO (VISION) | GEMINI 1.5 PRO | 3.53 | 0.413 | 0.0901 | 10.35 |
| *Ablation on Reasoning Component (LLM)* | | | | | |
| ⇄ GPT-4V | LLAMA3-70B | 3.55 | 0.411 | 0.0915 | 10.87 |
| ⇄ GPT-4V | QWEN2-72B | 3.57 | 0.409 | 0.0921 | 11.05 |
| *Ablation on Perception Component (VLM)* | | | | | |
| 📷 QWEN2.5-VL-MAX | GPT-4 TURBO | 3.59 | 0.408 | 0.0928 | 11.42 |
| 📷 LLAVA-NEXT-34B | GPT-4 TURBO | 3.64 | 0.403 | 0.0935 | 12.18 |
| *Performance of Full Open-Source Configurations* | | | | | |
| ☁ QWEN2.5-VL-MAX | LLAMA3-70B | 3.63 | 0.404 | 0.0931 | 12.01 |
| ☁ VIDEO-LLAVA-7B | LLAMA3-70B | 3.69 | 0.400 | 0.0940 | 12.96 |
| *Necessity of Semantic Perception Stage* | | | | | |
| ⊖ *N/A (Text Description)* | GPT-4 TURBO | 3.75 | 0.395 | 0.0948 | 18.52 |

## 4.3 ENHANCING OUT-OF-DISTRIBUTION GENERALIZATION

To rigorously evaluate the capability of `PRISM` in out-of-distribution (OOD) generalization, we conduct a specialized experiment on the SEVIR dataset, with results presented in Table 3. The experiment tests models trained on common storm types against a challenging OOD test set composed of rare and extreme weather events. The results compellingly demonstrate `PRISM`'s dual advantages. First, in terms of **absolute performance**, models augmented with `PRISM` consistently outperform their original counterparts in both ID and OOD scenarios, achieving significantly higher CSI scores when facing unseen extreme events. More critically, `PRISM` substantially enhances model **robustness** by mitigating performance degradation. For instance, while the original SimVP suffers a severe 26.2% drop in CSI when moving from ID to OOD data, the SimVP + `PRISM` configuration cuts this degradation by nearly half, to just **13.7%**. This marked reduction in performance decay provides unequivocal evidence that `PRISM`'s causal reasoning mechanism translates directly into superior generalization, ensuring more reliable predictions for challenging and unforeseen phenomena.

Table 3: **Out-of-Distribution Generalization on the SEVIR Dataset.** Performance comparison on In-Distribution (ID) and Out-of-Distribution (OOD) test sets. The ΔCSI column quantifies the percentage drop in CSI from ID to OOD performance, highlighting model robustness.

| MODEL | IN-DISTRIBUTION (ID) | | OUT-OF-DISTRIBUTION (OOD) | | DEGRADATION |
|---|---|---|---|---|---|
| | MSE ↓ | CSI ↑ | MSE ↓ | CSI ↑ | ΔCSI (% ↓) |
| SWIN (ORI) | 3.81 | 0.392 | 5.25 | 0.285 | 27.3 |
| SWIN + PRISM | **3.48** | **0.417** | **4.10** | **0.355** | **14.9** |
| SIMVP (ORI) | 3.52 | 0.447 | 4.98 | 0.330 | 26.2 |
| SIMVP + PRISM | **3.49** | **0.461** | **4.05** | **0.398** | **13.7** |
| FNO (ORI) | 4.50 | 0.310 | 6.80 | 0.190 | 38.7 |
| FNO + PRISM | **4.20** | **0.345** | **5.50** | **0.265** | **23.2** |

## 4.4 CASE STUDY: FORECASTING NAVIER-STOKES VORTICITY EVOLUTION

To illustrate the internal workflow of `PRISM`, we present a case study on the Navier-Stokes task in Figure 3. Our approach hinges on structured prompt engineering: we first guide a VLM to perform an expert-level interpretation of vorticity snapshots, extracting key physical events like vortex merging. Subsequently, these visual observations are combined with an explicit, text-explained

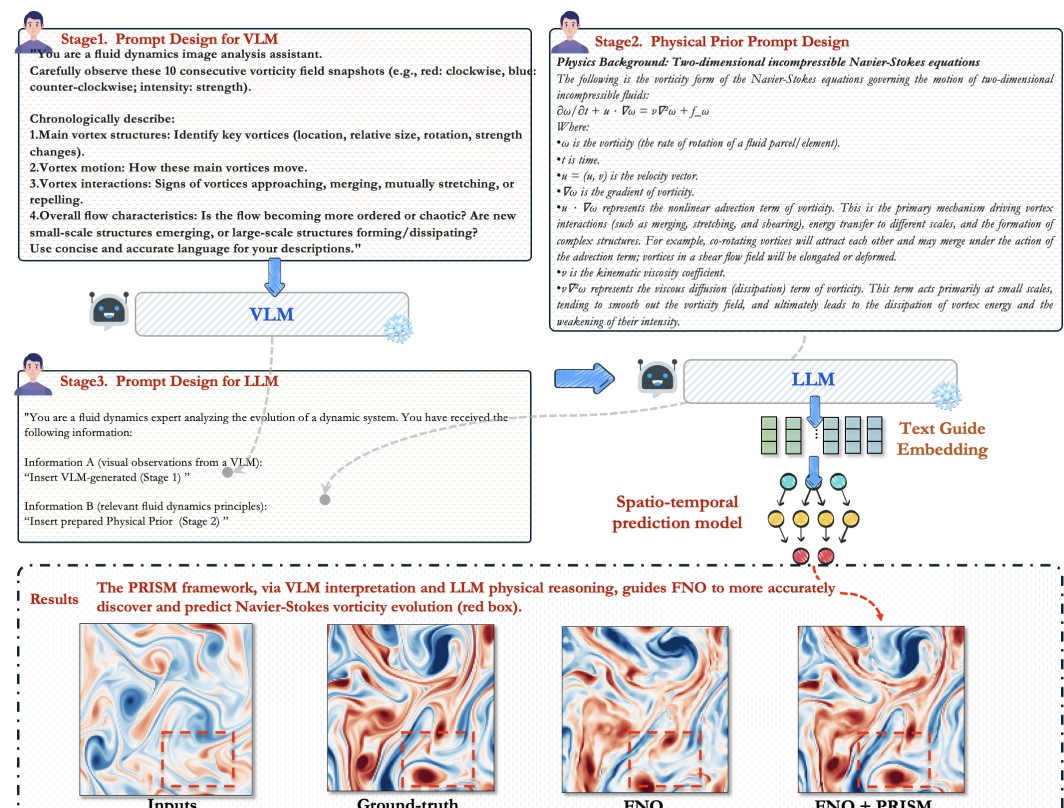

Figure 3: **The PRISM framework enhances physical process discovery in Navier-Stokes flow.**
(Top) PRISM pipeline: A VLM interprets vorticity inputs; an LLM, using physical priors like the
NS equations, generates textual guidance to empower a downstream prediction model (e.g., FNO).
(Bottom) Results: In Navier-Stokes prediction, compared to the input (first left) and ground-truth
(second left), the original FNO (second right) poorly predicts vorticity evolution in the red-boxed area.
With PRISM integration (FNO + PRISM, first right), predictions, informed by VLM-LLM physical
insights, are significantly improved and closer to the true process.

physical principle (the Navier-Stokes equations) and fed to an LLM. This "phenomenon + principle"
paradigm generates a causally-informed guiding text. The results compellingly demonstrate the
effectiveness of this approach. As shown in the visual comparison, the standard FNO prediction is
overly smooth and loses critical fine-grained structures. In stark contrast, FNO + PRISM successfully
reconstructs these high-frequency details with high fidelity to the ground truth, proving that our
method effectively compensates for the deficiencies of numerical models by injecting symbolic causal
knowledge, thereby enhancing physical realism and accuracy.

## 5 CONCLUSION

In this paper, we introduced PRISM, an innovative framework to address the poor generalization of
traditional data-driven forecasting models by rectifying their tendency to learn spurious correlations.
PRISM performs a novel causal intervention: it leverages a Vision-Language Model to decode raw
observations into rich semantic narratives, and then employs a Large Language Model to perform
causal reasoning with these narratives and explicit physical principles. This process generates a
symbolic guiding text that steers a downstream numerical predictor. Extensive experiments across
fluid dynamics, meteorology, and urban traffic demonstrate that PRISM, as a plug-and-play module,
significantly enhances model performance, especially in critical challenges like OOD generalization
and extreme event forecasting. Our work pioneers a new paradigm for "AI for Science" by *unifying
the powerful pattern recognition of traditional models with the superior symbolic reasoning of large
foundation models*, paving the way for more reliable and generalizable AI systems in science.

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

## A  THE USE OF LARGE LANGUAGE MODELS (LLMS)

LLMs were not involved in the research ideation or the writing of this paper.

## B  REPRODUCIBILITY STATEMENT

To ensure the reproducibility of our research, we provide a detailed description of our methodology in Section 3 and the experimental setup in Section 4. This includes specifics on datasets, evaluation metrics, and implementation details for all models. Our source code will be made publicly available upon the acceptance of this paper to facilitate verification and future research.

## C  TRAINING ALGORITHM

The training process of the `PRISM` framework follows a parameter-efficient paradigm. The large foundation models (VLM, LLM, and the text encoder) remain frozen, and only the parameters $\phi$ of the downstream spatio-temporal model, including its data encoder, fusion module, and decoder, are updated. Algorithm 1 details this procedure.

---

**Algorithm 1** `PRISM` Framework Training Algorithm

---

**Require:** Training dataset $\mathcal{D} = \{(\mathbf{X}_{\text{hist}}^{(P)}, \mathbf{X}_{\text{future}}^{(K)})\}$;
  1: Frozen foundation models: $\mathcal{M}_{\text{VLM}}, \mathcal{M}_{\text{LLM}}, \mathcal{E}_{\text{text}}$;
  2: Physical principles repository $\mathcal{P}_{\text{phys}}$;
  3: Number of training epochs $N_{\text{epochs}}$.
**Ensure:** Trained parameters $\phi$ of the downstream spatio-temporal model $f_\phi^*$.
  4: Initialize the trainable parameters $\phi$.
  5: **for** epoch = 1 to $N_{\text{epochs}}$ **do**
  6:     **for** each batch $(\mathbf{X}_{\text{hist}}^{(P)}, \mathbf{X}_{\text{future}}^{(K)})$ in $\mathcal{D}$ **do**
  7:                                   ▷ **Stage 1: Semantic Perception**
  8:         Render numerical grids $\mathbf{X}_{\text{hist}}^{(P)}$ into an image sequence $\mathbf{I}_{\text{hist}}$ using $\mathcal{V}$.
  9:         Generate the semantic narrative: $T_{\text{sem}} \leftarrow \mathcal{M}_{\text{VLM}}(\Pi_{\text{sem}}(\mathbf{I}_{\text{hist}}))$.
10:                                 ▷ **Stage 2: Causal Reasoning**
11:         Generate the causally-informed guidance text: $T_{\text{guide}} \leftarrow \mathcal{M}_{\text{LLM}}(\Pi_{\text{reason}}(T_{\text{sem}}, \mathcal{P}_{\text{phys}}))$.
12:                 ▷ **Stage 3: Guided Spatio-Temporal Prediction**
13:         Encode the guidance text into a dense vector: $\mathbf{g} \leftarrow \mathcal{E}_{\text{text}}(T_{\text{guide}})$.
14:         Encode historical observations into features: $\mathbf{H}_{\text{hist}} \leftarrow \mathcal{E}_{\text{data}}(\mathbf{X}_{\text{hist}}^{(P)})$.
15:         Fuse guidance and numerical features via attention: $\mathbf{c}_{\text{guided}} \leftarrow \mathcal{A}_{\text{guide}}(\text{Query} = \mathbf{g}, \text{Keys} = \mathbf{H}_{\text{hist}}, \text{Values} = \mathbf{H}_{\text{hist}})$.
16:         Generate the future prediction: $\hat{\mathbf{X}}_{\text{future}}^{(K)} \leftarrow \mathcal{D}_\phi(\mathbf{H}_{\text{hist}}, \mathbf{c}_{\text{guided}})$.
17:                         ▷ **Loss Calculation and Optimization**
18:         Calculate the prediction loss: $L \leftarrow \mathcal{L}(\hat{\mathbf{X}}_{\text{future}}^{(K)}, \mathbf{X}_{\text{future}}^{(K)})$.
19:         Update the trainable parameters $\phi$ by performing backpropagation on $L$.
20:     **end for**
21: **end for**
22: **return** Trained parameters $\phi$.

---

## D  EVALUATION METRICS

**Mean Squared Error (MSE)** measures the average of the squares of the errors—that is, the average squared difference between the estimated values and the actual value. It is one of the most common metrics for regression tasks. The formula is:

$$\text{MSE} = \frac{1}{n} \sum_{i=1}^{n} (Y_i - \hat{Y}_i)^2 \tag{8}$$

where $n$ is the number of data points (e.g., pixels), $Y_i$ is the observed (actual) value, and $\hat{Y}_i$ is the predicted value.

**Structural Similarity Index (SSIM)** is a perceptual metric that quantifies the visual quality degradation of a predicted image compared to a reference image. It considers changes in luminance, contrast, and structure. The formula is:

$$\text{SSIM}(x, y) = \frac{(2\mu_x\mu_y + C_1)(2\sigma_{xy} + C_2)}{(\mu_x^2 + \mu_y^2 + C_1)(\sigma_x^2 + \sigma_y^2 + C_2)} \tag{9}$$

where $x$ and $y$ are the two image windows being compared; $\mu_x$ and $\mu_y$ are the pixel sample means of $x$ and $y$; $\sigma_x^2$ and $\sigma_y^2$ are the variances of $x$ and $y$; $\sigma_{xy}$ is the covariance of $x$ and $y$; and $C_1, C_2$ are stabilization constants to avoid division by a small denominator.

**Mean Absolute Percentage Error (MAPE)** measures the average magnitude of errors as a percentage of the actual values. It is often used for assessing the relative error of a forecast. The formula is:

$$\text{MAPE} = \frac{100\%}{n} \sum_{i=1}^{n} \left| \frac{Y_i - \hat{Y}_i}{Y_i} \right| \tag{10}$$

where $Y_i$ is the actual value and $\hat{Y}_i$ is the forecast value. This metric is undefined when any actual value $Y_i$ is zero.

**Critical Success Index (CSI)**, also known as the Threat Score (TS), is used for evaluating the performance of categorical forecasts (e.g., predicting storm events). It measures the fraction of correctly predicted events out of all predicted or actual events. The formula is based on the components of a confusion matrix:

$$\text{CSI} = \frac{\text{TP}}{\text{TP} + \text{FN} + \text{FP}} \tag{11}$$

where:

- **TP (True Positives)**: The number of correctly predicted events (Hits).
- **FN (False Negatives)**: The number of events that occurred but were not predicted (Misses).
- **FP (False Positives)**: The number of predicted events that did not occur (False Alarms).

# E  NOTATION

Table 4 provides a comprehensive summary of the key notations used throughout our methodology.

Table 4: Summary of notations used in the methodology.

| Symbol | Description |
|---|---|
| ***System State and Observations*** | |
| $\mathcal{S}, \mathcal{X}$ | State space and Observation space of the dynamical system, respectively. |
| $\mathbf{s}_t \in \mathcal{S}$ | The true, latent state of the system at time $t$. |
| $\mathbf{x}_t \in \mathcal{X}$ | The observation of the system at time $t$. |
| $P, K$ | Length of the historical lookback window and the future forecast horizon. |
| $\mathbf{X}_{\text{hist}}^{(P)}$ | Sequence of $P$ historical observations $(\mathbf{x}_{t-P+1}, \ldots, \mathbf{x}_t)$. |
| $\mathbf{X}_{\text{future}}^{(K)}$ | Sequence of $K$ future observations to be predicted $(\mathbf{x}_{t+1}, \ldots, \mathbf{x}_{t+K})$. |
| $\hat{\mathbf{X}}_{\text{future}}^{(K)}$ | The model's predicted sequence of future observations. |
| $S_t, \mathcal{P}_{\text{phys}}$ | Abstract latent semantic structures and invariant physical principles. |
| $\mathcal{D}$ | The underlying data distribution. |
| ***Model Components and Parameters*** | |
| $\mathcal{M}_{\text{VLM}}$ | The Vision-Language Model (VLM) used for semantic perception. |
| $\mathcal{M}_{\text{LLM}}$ | The Large Language Model (LLM) used for causal reasoning. |
| $\mathcal{E}_{\text{text}}$ | The pre-trained text encoder for generating guidance vectors. |
| $\mathcal{E}_{\text{data}}$ | The data encoder for processing numerical observations. |
| $\mathcal{A}_{\text{guide}}$ | The Guidance-Attention Module for multi-modal fusion. |
| $\mathcal{D}_\phi$ | The decoder of the downstream spatio-temporal model. |
| $f_\theta, f_\phi^*$ | Conventional prediction model and the `PRISM`-enhanced model. |
| $\theta, \phi$ | Learnable parameters for the conventional and enhanced models. |
| $\mathbf{W}_Q, \mathbf{W}_K, \mathbf{W}_V$ | Learnable projection matrices for attention (Query, Key, Value). |
| ***Intermediate Representations*** | |
| $\mathcal{T}, \mathcal{I}$ | The text space and image space, respectively. |
| $T_{\text{sem}}$ | The semantic narrative (text) generated by the VLM. |
| $T_{\text{guide}}$ | The final, causally-informed guidance text from the LLM. |
| $\mathbf{g} \in \mathbb{R}^{d_g}$ | The dense guidance vector encoded from $T_{\text{guide}}$. |
| $\mathbf{H}_{\text{hist}}$ | Sequence of numerical features $(\mathbf{h}_1, \ldots, \mathbf{h}_P)$ from $\mathcal{E}_{\text{data}}$. |
| $\mathbf{c}_{\text{guided}}$ | The guided context representation from the attention module. |
| $\alpha_i$ | The attention weight for the $i$-th historical timestep. |
| ***Operators and Functions*** | |
| $\mathcal{O}$ | The observation operator mapping true states $\mathbf{s}_t$ to observations $\mathbf{x}_t$. |
| $\mathcal{G}$ | The complete process for generating the guidance variable $\mathbf{g}$. |
| $\Psi_{\text{sem}}$ | The semantic perception operator (Stage 1). |
| $\Psi_{\text{reason}}$ | The causal reasoning operator (Stage 2). |
| $\mathcal{V}$ | The visualization map from numerical grids to images. |
| $\Pi_{\text{sem}}, \Pi_{\text{reason}}$ | Prompting functions for the VLM and LLM, respectively. |
| $\mathcal{L}$ | The loss function used for model training. |

