# OpenReview forum: "CAUSAL REASONING WITH LARGE FOUNDATION MOD- ELS TO GUIDE DYNAMIC SYSTEM FORECASTING"
_ICLR.cc/2026/Conference — ICLR 2026 Conference Withdrawn Submission_

### Official Review · Reviewer_ZMER · 2025-10-23

**Soundness:** 3
**Presentation:** 3
**Contribution:** 2
**Rating:** 4
**Confidence:** 5

**Summary:**

This paper studies the spatio-temporal forecasting problem. The authors propose the Physics-informed Reasoning and Interpretation for Spatio-temporal Modeling (PRISM) method to address the spurious correlations problem. Specifically, PRISM utilizes the vision-language model (VLM) to process the spatial-temporal snapshots into semantic narratives, then an LLM will reason through the narratives to generate the causal textual guidance. Authors conduct experiments on several real-world applications (e.g., weather forecasting) to show the effectiveness of PRISM.

**Strengths:**

1. The motivation is sound and clearly demonstrated. I agree with the authors that the spatial-temporal modeling problem is important and challenging, and pure data-driven methods may lack the ability to capture causal or physical mechanisms.

2. Figure 2 is well drawn, clearly showing the procedure of the proposed method.

3. The main experiments and ablation studies are comprehensive, validating the authors' claims in the introduction.

**Weaknesses:**

1. I suggest that authors revise Figure 1 (a). I believe the current version cannot truly reflect the spurious association. In causal inference, the spurious association is mainly represented by a common cause [1], rather than just an arrowhead or a shortcut. Besides, I think there exists a causal relationship between the historical information and true prediction (rather than pure spurious association).

2. Since the spatial-temporal modeling is an active and important area and this work is inspired by causality. I would encourage authors to discuss more works of spatial-temporal causal inference, which include (but are not limited to): [2] and [3].

3. I believe authors didn't discuss one of the most important works of MLLMs: LLaVA (visual instruction tuning) [4].

4. How did authors handle the hallucination problem of MLLMs or LLMs? I believe the current MLLMs may exhibit some serious hallucination problems. What if the outputs of MLLMs are incorrect or not accurate?

5. I am not totally convinced by the authors' implementation of the prediction. It seems that the only causal (physical) part is the generation of the guidance text; the (most) important prediction part seems to be the simple fusion of text and historical features. I suggest that authors elaborate on this part.

> [1] Pearl J. Causal inference in statistics: An overview[J]. 2009.

> [2] Li H, Chi H, Liu M, et al. Transformer-Based Spatial-Temporal Counterfactual Outcomes Estimation[C]//Forty-second International Conference on Machine Learning.

> [3] Christiansen, R., Baumann, M., Kuemmerle, T., Mahecha, M. D., and Peters, J. Toward causal inference for spatio-temporal data: conflict and forest loss in colombia. Journal of the American Statistical Association, 117(538): 591–601, 2022.

> [4] Liu H, Li C, Wu Q, et al. Visual instruction tuning[J]. Advances in neural information processing systems, 2023, 36: 34892-34916.

**Questions:**

Please refer to the weakness part.

---

### Official Review · Reviewer_dcVx · 2025-10-23

**Soundness:** 2
**Presentation:** 1
**Contribution:** 2
**Rating:** 2
**Confidence:** 3

**Summary:**

The paper introduces PRISM, a wrapper around a spatio-temporal predictor for short-horizon forecasting. The task can be described as follows: given historical grid frames, predict future frames on tasks like traffic flow, convective weather nowcasting, and 2D Navier-Stokes vorticity. PRISM uses a frozen VLM->LLM pipeline to convert recent history into text, combine it with textual "physical priors" and produce a guidance text. The text is embedded by a frozen encoder into a single vector $g$. A trainable cross-attention module uses $g$ as a query over the historical features that produces a context that is used to condition the decoder to forecast the future. Only the fusion system and the downstream predictor are trained. The system is evaluated on multiple model backbones.

**Strengths:**

- The paper tackles the important problem of dynamical systems forecasting.
- Physically grounding the forecasting process is a sensible direction.
- Evaluated across a broad set of backbone models.
- Modular design, easy to graft to many backbones.
- There are sizeable improvements on the evaluated tasks.

**Weaknesses:**

The formalization, as presented, introduces ambiguity in several places:
1) VLMs and LLMs are treated as deterministic operators.
2) LLMs consume ordered token sequences, not sets.
3) "Causal intervention" is rhetorical in the math. Eq 3 has no causal graph, no do operators, no invariances, no identification results.
4) The LLM does not generate a probabilistic description of future states as claimed. In practice, the authors sample one $T_{\text{guide}}$ and embed it.
5) Attention weights are not faithful importance measures without conditions (Wiegreffe & Pinter, EMNLP-IJCNLP 2019; Serrano & Smith, ACL 2019; Bastings & Filippova, BlackboxNLP 2020; inter alia)

Furthermore:
- The methodology is unnecessarily complicated and hard to follow in its current state. Types (domains/codomains) for functions and variables are missing, obscuring their roles.
- The manuscript uses promotional phrasing instead of neutral reporting (e.g., "remarkable reduction", "chart-topping 0.089", "A Pioneering Paradigm").
- Overstated claims (examples):
L101 "Fundamentally enhancing its generalization and reliability,"
L319 "universal applicability,"
L411 "unequivocal evidence,"
L405 "The results compellingly demonstrate,"
L470 "proving that our method effectively compensates...".
- The paper uses causal terminology, but the setup is not causal.
- Limited ablations on parts of the system, see questions.
- Some experimental details are missing, see questions.
- The linked repository appears to be a placeholder rather than complete, runnable code.

**Questions:**

Questions/Remarks

- Please consider removing causal claims beyond motivation. Describe it as text-guided conditioning instead.
- Simplify the methodology. Consider dropping the "reasoning prompt function" over the powerset of text sets: this is just feeding the VLM summary into an LLM with a fixed prompt.
- How do we sample relevant physical priors from the repository of physical knowledge? Specify the repository per domain (verbatim items), how priors are selected (rule or heuristic), their ordering, and token budgets. Include a control with shuffled/irrelevant priors.
- Claim at L080–L084 (strong statement on the fundamental reason of OOD failures). Either cite supporting work or demonstrate it empirically in this paper.
- Please replace overstatements with precise, statistics-backed statements and soften causal/priority claims to match the presented evidence.
- Ablations:
1) Skip the LLM path and use the VLM in its stead to investigate the need for the LLM.
2) Use the LLM’s hidden representations (i.e., last hidden state) instead of decoding to language and then re-encoding where possible (open-weights models).
3) Experiment with different fusion methods such as concatenation
4) Experiment with multi-query vs single pooled vector ($g$).

(You don’t have to run all of these. Pick the most informative.)

- Table 1 contains counterexamples to the headline claims: UNO degrades on Navier Stokes and FNO doesn’t change at all. Analysis/explanation is needed.
- The OOD split in experiments is described narratively, not formally. Define the OOD set precisely, explain why it is challenging.
- Tuning parity: Document hyperparameters/search budgets and confirm parity with and without PRISM for each backbone.
- The paper states ten backbones, but Table 1 enumerates nine (ResNet, U-Net, ViT, SWIN, SimVP, PastNet, FNO, CNO, UNO). Please either add the missing 10th model with results and training details, or correct the count to nine.

---

### Official Review · Reviewer_7y1c · 2025-10-26

**Soundness:** 2
**Presentation:** 3
**Contribution:** 2
**Rating:** 4
**Confidence:** 3

**Summary:**

This paper proposes PRISM, a causal intervention framework that uses a VLM to extract semantic narratives from spatiotemporal observations and a LLM to incorporate physical principles into textual guidance that steers downstream predictors. Experiments demonstrate consistent performance gains across fluid dynamics, weather, and urban traffic forecasting tasks, especially in OOD and extreme-event scenarios.

**Strengths:**

1. The plug-and-play PRISM module yields consistent improvements across 10 architectures and 3 domains, showing usefulness beyond a single model.

2. The OOD results show substantial reduction in degradation, supporting claims of better generalization.

**Weaknesses:**

1.The causal argument is mainly motivational; there is no empirical test of causal variables or interventions beyond better metrics, making the “repair causal pathway” claim not fully justified.

2. The generated semantic/physical guidance is not checked against ground truth expert physical reasoning, so improvements may come from generic feature enrichment rather than true causal reasoning.

3. Prompting for VLM/LLM is critical (e.g., physical principles injected as text), but the paper omits prompt templates and reasoning examples, hurting reproducibility. The provided repository is empty.

**Questions:**

1. Table 2 suggests performance scales with model capacity; could gains be primarily caused by richer text features rather than incorporation of physical laws, and can authors isolate this via prompts without physics input.


2. Figure 3 shows guidance improves vortex prediction, but what evidence confirms the LLM actually reasons with Navier-Stokes equations, rather than hallucinating plausible phrases.


3. Table 1 includes some negative improvement cases (e.g., ViT on TaxiBJ+ is +0.9% MSE worse); what conditions cause failure, and how does this align with the causal story.


4. Guidance text length/quality could affect performance; did authors analyze whether longer guidance vs higher-quality semantics correlates better with improvement.

---

### Official Review · Reviewer_Xipu · 2025-11-04

**Soundness:** 3
**Presentation:** 3
**Contribution:** 2
**Rating:** 4
**Confidence:** 3

**Summary:**

This paper introduces PRISM, a framework that addresses fundamental limitations in data-driven spatio-temporal forecasting models. The authors identify that traditional models learn spurious correlations by bypassing latent semantic states and physical principles. Extensive experiments demonstrate that PRISM consistently improves diverse architectures as a plug-and-play module, achieving substantial gains in OOD generalization, enhanced prediction accuracy, and improved extreme event forecasting. The work pioneers a new paradigm unifying traditional pattern recognition with symbolic causal reasoning from large foundation models.

**Strengths:**

- The paper convincingly argues for a new AI for Science paradigm where foundation models serve as reasoning partners rather than replacements for numerical methods.
- The core insight that data-driven spatio-temporal models fail because they learn spurious correlations while bypassing latent semantic states and physical principles is a fresh perspective.

**Weaknesses:**

- The paper frames its contribution as "causal intervention" but the evidence provided doesn't adequately support true causal mechanisms. The VLM-LLM pipeline generates descriptions and predictions about physical phenomena, but there's no evidence these capture actual causal pathways rather than sophisticated correlations. To substantiate causal claims, the paper should include intervention experiments. What happens if you deliberately inject incorrect physical principles or contradictory semantic descriptions? If the system is truly learning causal pathways, performance should degrade predictably.
- The paper claims to bridge semantic and physics gaps but doesn't compare against physics-informed methods, such as PINNs with proper PDE loss terms.
- The abstract and introduction claim PRISM "improves prediction under data sparsity" but this is barely evaluated.
- Some similar works are not discussed.

[1] Unveiling causal reasoning in large language models: Reality or mirage? NeurIPS, 2024.
[2] Is knowledge all large language models needed for causal reasoning?

**Questions:**

- The motivation is not clear. They claimed that traditional models learn a spurious correlation by ignoring latent semantic and physical factors. Physical factors are clear. What is the definition of semantic factors here? They'd better explain it with a practical example. In my view, LLMs have semantic understanding capability.
-  Running GPT-4V + GPT-4 Turbo for every training sample is likely expensive. How many API calls were made? What's the latency? For the TaxiBJ+ dataset, with presumably thousands of training samples, this could be prohibitive.
-  Do VLMs genuinely extract semantic concepts or just perform sophisticated pattern matching? For example, does GPT-4V identify "vortex merging" because it understands fluid dynamics or because it pattern-matches visual features to text it was trained on?

---

### Comment · Area_Chair_7DR5 · 2025-11-27

Dear Authors and Reviewers,

The discussion phase will end soon. If you want to further discuss comments and replies with each other, please post your thoughts by adding official comments.

Thanks for your efforts and contributions to ICLR 2026.

Best regards,

Your Area Chair

---

### Note · Authors · 2026-01-06

I have read and agree with the venue's withdrawal policy on behalf of myself and my co-authors.